# Effect of COVID-19-Related Lockdown οn Hospital Admissions for Asthma and COPD Exacerbations: Associations with Air Pollution and Patient Characteristics

**DOI:** 10.3390/jpm11090867

**Published:** 2021-08-30

**Authors:** Ioanna Sigala, Timoleon Giannakas, Vassilis G. Giannakoulis, Efthimios Zervas, Aikaterini Brinia, Niki Gianiou, Andreas Asimakos, Efi Dima, Ioannis Kalomenidis, Paraskevi Katsaounou

**Affiliations:** 1Pulmonary and Respiratory Failure Department, First ICU, Evangelismos Hospital, Ipsilandou 45-7, 10676 Athens, Greece; giannasig@yahoo.com (I.S.); nikigianniou@yahoo.com (N.G.); a.asimakos@yahoo.gr (A.A.); efi_dima@yahoo.gr (E.D.); ikalom@med.uoa.gr (I.K.); 2Medical School, National Kapodistrian University of Athens, 10679 Athens, Greece; tgiannakas@gmail.com (T.G.); giannakoulisv@gmail.com (V.G.G.); kbrinia@yahoo.gr (A.B.); 3Laboratory of Technology and Policy of Energy and Environment, School of Science and Technology, Hellenic Open University, Parodos Aristotelous 18, 26335 Patra, Greece; zervas@eap.gr

**Keywords:** COVID-19, SARS-SoV-2, hospital admissions, COPD exacerbation, asthma exacerbation, air pollution

## Abstract

We conducted a retrospective observational study to assess the hospitalization rates for acute exacerbations of asthma and COPD (chronic obstructive pulmonary disease) during the first imposed lockdown in Athens, Greece. Patient characteristics and the concentration of eight air pollutants [namely, NO (nitrogen monoxide), NO_2_ (nitrogen dioxide), CO (carbon monoxide), PM2.5 (particulate matter 2.5), PM10 (particulate matter 10), O_3_ (ozone), SO_2_ (sulfur dioxide) and benzene] were considered. A total of 153 consecutive hospital admissions were studied. Reduced admissions occurred in the Lockdown period compared to the Pre-lockdown 2020 (*p <* 0.001) or the Control 2019 (*p* = 0.007) period. Furthermore, the concentration of 6/8 air pollutants positively correlated with weekly hospital admissions in 2020 and significantly decreased during the lockdown. Finally, admitted patients for asthma exacerbation during the lockdown were younger (*p* = 0.046) and less frequently presented respiratory failure (*p* = 0.038), whereas patients with COPD presented higher blood eosinophil percentage (*p* = 0.017) and count (*p* = 0.012). Overall, admissions for asthma and COPD exacerbations decreased during the lockdown. This might be partially explained by reduction of air pollution during this period while medical care avoidance behavior, especially among elderly patients cannot be excluded. Our findings aid in understanding the untold impact of the pandemic on diseases beyond COVID-19, focusing on patients with obstructive diseases.

## 1. Introduction

Both asthma and chronic obstructive pulmonary disease (COPD) exacerbations are notable causes of death globally, with viral infections, air pollution, chemical irritants and stress as common major precipitators [1,2,3,4,5]. These precipitators have been variably affected during the COVID-19 pandemic, possibly increasing hospital admissions of exacerbated patients. It is hinged on the premise that since SARS-CoV-2 is predominantly isolated in the respiratory tract, its increased transmission could be a possible viral trigger of asthma or COPD attacks [6,7]. Conversely, imposed lockdowns during the pandemic may have reduced asthma or COPD exacerbations, either as a true phenomenon, for example, due to reduced exposure to environmental triggers and air pollution [8], or due to medical care avoidance behavior [9].

Notwithstanding, the overall effect of the pandemic on asthma and COPD exacerbations remains unclear. Hence we aim to study the rate of asthma and COPD exacerbation admissions during the period of Lockdown in Athens, Greece. Admissions per week, patients’ characteristics, and air pollutant concentrations were collected and analyzed. We thus hypothesize that reduced patients’ hospitalization could be hinged on reduction in air pollutants from and reduced exposure to infection causes of exacerbations.

## 2. Materials and Methods

### 2.1. Design and Patient Population

We conducted a retrospective observational study to assess hospitalization rates for acute exacerbations of asthma and COPD in “Evangelismos” hospital during the first imposed lockdown in Greece. “Evangelismos” is the biggest tertiary hospital in Athens, Greece as well as a COVID-19 reference center. All patients admitted in the respiratory ward with a confirmed diagnosis of asthma/COPD attack for a period of 17 weeks (Monday to Sunday) in 2020 (3 February–31 May, 17 weeks) and 17 weeks in 2019 (4 February–2 June, 17 weeks) and had a negative Polymerase Chain Reaction (PCR) test for SARS-Cov-2 at the ED (emergency department), were included. We further aimed to link intra-year (2020) fluctuations in weekly admissions with patient characteristics as well as with air pollution levels. The study was conducted in accordance with the declaration of Helsinki and approved by The Institutional Review Board of “Evangelismos” (Protocol Number: 281/2020).

### 2.2. Study Periods

The first COVID-19 case in Greece was identified on 26 February 2020. On 11 March 2020 the Greek government announced the shutdown of schools and universities and recommended self-confinement behaviour. Therefore, we adopted 9th of March (Monday) 2020 as the beginning of the lockdown period and the quarantine spans through 8 weeks which terminated on 3 May—the date that drastic measures were relaxed. To perform comparative analyses between the COVID-19 study period (lockdown) and periods without exposure to COVID-19 as well as the time after COVID-19 lockdown, we created six study periods, namely:(1)Lockdown: 9 March–3 May 2020 (8 weeks, Monday to Sunday)(2)Control: 11 March–5 May 2019 (8 weeks, Monday to Sunday)(3)Pre-lockdown: 3 February–8 March 2020 (5 weeks, Monday to Sunday)(4)Pre-Control: 4 February–10 March 2019 (5 weeks, Monday to Sunday)(5)Post-lockdown: 4 May–31 May 2020 (4 weeks, Monday to Sunday)(6)Post-Control: 6 May–2 June 2019 (4 weeks, Monday to Sunday)

### 2.3. Data Collection

Patient demographic, clinical and laboratory data as well as admission dates and total length of stay were collected retrospectively via review of charts for the respective study periods of 2019 and 2020. The presence of lethargy/coma, respiratory failure, hypercapnia and intubation or death were utilized as markers of severity. Air pollution data were collected from the Greek Ministry of Environment and Energy. Numerous air quality monitoring stations were used in order to objectively quantify Attica’s air pollution. It was not possible to collect data from every station; therefore, we selected four air quality monitoring stations in representative regions of Attica (namely Elefsina, NeaSmirni, Patision street, and Piraeus). Mean daily air pollutants concentrations of eight air pollutants were used; nitrogen monoxide (NO), nitrogen dioxide (NO_2_), carbon monoxide (CO), particulate matter (PM2.5, PM10), ozone (O_3_), sulfur dioxide (SO_2_) and benzene. In order to facilitate comparisons, we calculated the average weekly concentrations for each of the 17 weeks of 2020 in our study (5 weeks for Pre-Lockdown, 8 for Lockdown, 4 for Post-Lockdown). Not all air pollutants were measured in each monitoring station. As a result, PM2.5, O_3_, CO, SO_2_ are the average value of two stations, PM10 of three, while data for NO, NO_2_, and benzene were available in all four stations.

### 2.4. Primary and Secondary Outcomes

The primary outcome of the study was asthma and/or COPD hospitalization incidence rate (IR) expressed as admissions per week. The IR between the Lockdown and each of the remaining study periods were compared through Incidence Rate Ratios (IRR).

Secondary outcomes were: (a) Intra-year (2020) correlations between air pollutant concentrations and hospital admissions and (b) differences in population characteristics and air pollutant concentrations between the Pre-lockdown and Lockdown periods.

### 2.5. Statistical Analysis

Crude weekly IR for asthma/COPD attack admissions were calculated by dividing the number of cumulative admissions by the number of weeks for each study period. IRR with 95% Confidence Intervals (CI) were calculated between each study period and the Lockdown using Poisson regression (log-linear model). Categorical variables are presented as counts and percentages and were compared using the chi-square or Fisher’s exact test as appropriate. Continuous variables included in the analysis are presented as median (interquartile range); comparisons were made using the Mann-Whitney U test. Correlations between hospital admissions per week and the respective average weekly air pollutant concentrations were performed through Spearmans’ rank correlation coefficient (r). Correlations are additionally presented as scatterplots at Appendix A). A *p*-value equal to or below 0.05 denoted statistical significance. All analyses were done with IBM SPSS Statistics 26.0.

## 3. Results

A total of 153 hospital admissions for asthma or COPD exacerbations were included in this study. Characteristics of patients’ admissions and outcomes, categorized according to the respective disease are presented in Table 1. Sixty-eight patients were admitted for asthma attack and 83 patients (85 admissions as 2 patients were admitted twice in different time frames) for COPD exacerbation. The vast majority of patients had a preexisting diagnosis of asthma (93%) or COPD (86%), while few patients were undiagnosed at the time admission. As presented in Table 1, asthma and COPD patients differed significantly in the majority of parameters studied. Specifically, compared to asthmatics, COPD patients were older, male predominant, more frequently active smokers, with more severe disease and with longer hospitalization. Additionally, asthma patients complained less frequently about sputum, had higher eosinophil count and lower serum CRP value at presentation.

### 3.1. Primary Outcome

#### 3.1.1. IR of Asthma and COPD Exacerbations

Asthma and COPD hospitalization IR is presented in Table 2. Specifically, the IR of asthma attack admissions in the lockdown period (IR 0.625) was significantly lower when compared to the Pre-lockdown period (IR 2.8; IRR = 4.48, *p* = 0.004) as well as the Pre-Control (IR 2; IRR = 3.2, *p* = 0.034), Control (IR 1.875; IRR = 3, *p* = 0.033) and Post-Control (IR 4.5; IRR = 7.2, *p <* 0.001) periods. The IR rate of COPD exacerbation admissions in the lockdown period (IR 1.375) was significantly lower when compared to the Pre-lockdown (IR 3.4; IRR = 2.5, *p* = 0.019) and Pre-control period (IR 3.2; IRR = 2.3, *p* = 0.031) but not when compared to the Post- lockdown, Control and Post-Control periods. Figure 1a,b graphically depict the cumulative sum of admissions per week during the years 2020 and 2019, for asthma and COPD respectively.

#### 3.1.2. IR of Total Hospital Admission Rates

The IR of obstructive lung disease (asthma and COPD) admissions in the Lockdown period (IR 2) was significantly lower when compared to the one observed during the Pre- Lockdown (IR 6.2; IRR = 3.1, *p <* 0.001), Post- Lockdown (IR 4; IRR = 2, *p* = 0.05), Pre-Control (IR 5.2; IRR = 2.6, *p* = 0.003), Control (IR 4.5; IRR = 2.3, *p* = 0.007) and Post-Control (IR 7; IRR = 3.5, *p <* 0.001) periods. Figure 1c graphically depicts the cumulative sum of admissions per week during 2020 and 2019.

### 3.2. Secondary Outcomes

#### Patient Population Characteristics: Lockdown versus Pre-Lockdown

We aim to identify differences in the baseline characteristics between patients admitted during the Pre- Lockdown and Lockdown periods. As baseline population characteristics between the diseases of interest (asthma and COPD) significantly differed (Table 1), we performed 2 separate comparisons, one for each disease. As presented in Table 3, asthma patients during the Lockdown period were of younger age, whereas there was no difference in age between different periods for COPD patients. COPD patients during the Lockdown exhibited higher blood eosinophil count (170 vs. 10, *p* = 0.012) and percentage (2.2% vs. 0.1%, *p* = 0.017). In terms of disease severity, fewer patients admitted for asthma exacerbation during the Lockdown period presented with respiratory failure at admission when compared to patients from the Pre-Lockdown period (20% vs. 78.6%, *p* = 0.038). Although not meeting statistical significance due to the small size of our cohort there was no intubation or death due to asthma attack at the Lockdown period (0% vs. 7.1% at the pre-Lockdown period). There was no difference in disease severity at presentation between COPD patients for the two respective periods. Finally, none of the patients in the Lockdown period tested positive for influenza.

Air pollutants concentrations in the Lockdown period vs the Pre- Lockdown period are depicted in Figure 2. A statistically significant reduction was observed in the air concentration of NO_2_ (*p* = 0.028), NO (*p* = 0.028), PM10 (*p* = 0.019), PM2.5 (*p* = 0.04), SO_2_ (*p* = 0.028), benzene (*p* = 0.005) and CO (*p* = 0.019), while that of O_3_ was significantly increased (*p* = 0.028).

Table 4 shows the Spearman’s correlation coefficients (r) between the concentration of air pollutants and the total number of asthma/COPD/exacerbations for 2020.The concentration of NO, NO_2_, PM2.5, CO, and benzene significantly correlated to the number of total admissions. Specifically, for each disease of interest, the concentration of PM2.5, CO, SO_2_ and benzene were significantly positively correlated with the number of asthma admissions per week and NO, NO_2_, PM2.5, CO and that of benzene with the number of COPD admissions per week. The concentration of ozone (O_3_) was significantly inversely related to the number of COPD and total admissions per week.

## 4. Discussion

This study demonstrates that hospital admissions for obstructive airway diseases (asthma and/or COPD) were significantly reduced in Athens, Greece during the first 2020 lockdown. Hospital admissions were most remarkably reduced on asthmatics than COPD and correlated with the reduction in air pollutants observed during the lockdown. Asthmatic exacerbations during the Lockdown as measured by the presence of respiratory failure, intubation or death were less severe than the pre- Lockdown period, and asthmatics admitted during the Lockdown were younger. COPD exacerbations didn’t differ in severity but COPD patients presented with more blood eosinophils at admission in the lockdown period, both in percentage and absolute number.

Arguably, the most meaningful comparisons were among the Lockdown period and the Pre-Lockdown (intra-year reduction; IRR = 3.1 for total admissions, *p <* 0.001) or the Control period (inter-year reduction; IRR = 2.3 for total admissions, *p* = 0.007). During the Lockdown, reduced air pollutants concentrations occurred (with the exception being O_3_), and we further exhibited an overall positive correlation of weekly hospital admissions with several air pollutants concentration (with the exception again being O_3_ which was inversely correlated), an observation that partially explains our primary finding.

There are reports about reduced emergency department visits and general hospital admissions during imposed lockdown for COVID-19 [10,11]. Exacerbation of obstructive airway diseases (COPD/Asthma) were among the diseases that exhibited consistently reduced hospitalization rates during the Lockdown. Concerning COPD, reports from Germany, Spain, Hong Kong, Singapore, and the USA present similar findings [10,12,13,14,15]. In the aforementioned studies, the reduction is attributed to universal masking, social distancing, and reduced air pollution [8]. Considering severe COPD exacerbations (i.e., COPD exacerbations with a need for hospitalization), studies from Singapore, Germany, China, USA etc. reported up to 60% reduction in hospitalization rate [10,12,14,15]. This reduction was attributed to universal masking and social distancing resulting in fewer viral infections [15]. Specifically, Tan et al. [15] reported a reduction in respiratory viral infections detection rate from 49 to 11%, whereas Chan et al. [14] reported a decreased incidence of Influenza A-B, which correlated with the reduction in COPD exacerbations. Although we cannot demonstrate a statistically significant reduction in influenza infections, none of the patients tested positive for influenza during the Lockdown. In our study, lockdown resulted in a 60% reduction in COPD admissions compared to the Pre-Lockdown period. Additionally to the previous factors that were analysed, COPD patients’ perception of vulnerability could have prompted a strict adherence to guidelines and good practices as suggested by improvement in asthma and COPD treatment adherence during the COVID-19 pandemic [16]. Lastly, we cannot exclude a contribution of fear of getting infected in the hospital and medical care avoidance behavior to the observed admission reduction [17].

The reduction of hospitalization has been noticed in the whole area of Greece in a multicentre study that our hospital also took part [18]. Contrastingly, our study demonstrated that reduced air pollutants correlated with the reduction in exacerbations and provide details form the demographics and patients’ hospital files.However in this study there were no comparison with the two previous years and there was nor data on pollution nor details from the patients demographics and history [18].

In our current study, COPD patients reveal higher eosinophils (counts and percentage) at admission, during Quarantine. This increase could be attributed to the reduction in infectious cause of exacerbations, although we cannot provide evidence for that and this could be a random finding. It has been revealed that in severe COPD exacerbations eosinophils at admission is associated with shorter length of stay [19,20,21,22], fewer early treatment failures [20], higher probability of non-infectious cause of exacerbation [21,23], or viral-dominant infections [24] and higher probability of relapse [20]. However, in our study there was neither a change in the duration of hospitalization nor in the severity of exacerbations (as measured by the presence of hypercapnia, intubation/death) during the lockdown, a finding similar to that of Chan et al. [14]. COPD exacerbations are mainly triggered by respiratory infections and air pollution. They accelerate lung function decline, worsen quality of life, and increase overall morbidity and mortality of COPD patients. Therefore, reduction of exacerbations is one of the major goals of COPD management. If the combination of reduced exposure to infection due to self-protection measures (social-distancing and use of face masks) and the decrease in air pollutants attributable to reduced human activities can result in a such a reduction of COPD exacerbation, this finding needs to be seriously considered as it may partially compensate for public health consequences of COVID-19 on chronic respiratory diseases [24].

Evidence about hospital admission in adult asthmatics during lockdown is scarce. In Massachusetts there was a 64% decline in asthma admission of both children and adults [11] and in Japan [25] a 54% reduction in adult admissions, whereas reports from Italy [26] and Saudi Arabia [27] from patients with mainly severe asthma suggest good control with no increase in exacerbations frequency. Namely, a survey in Italy, a neighboring country to Greece, with similar climate, reported a low number of asthma exacerbations and good disease control during the Quarantine, as specialists advocated drug adherence and furthermore, home isolation may have reduced late winter/spring pollen exposure [26]. The majority of reports refer to children demonstrating reduced admission [25,28,29,30], emergency department visits [31], health care visits [28] and less rhinovirus infection [28]. Although good asthma control during quarantine could be explained by reduced allergens exposure, better air quality, decreased viral infections and better adherence to medication [16]; medical care avoidance cannot be excluded.

In our study adult asthmatics that were hospitalized for severe exacerbations in Athens were remarkably less during the Lockdown and the Post-Lockdown period. Although a rebound increase in admission was observed at the Post-Lockdown period, hospitalizations stayed way beyond 2019 levels. Furthermore, asthmatics were younger with less severe exacerbation (measured as the presence of respiratory failure) at the Lockdown period. Indeed, in our study, respiratory failure was present in 20% of patients admitted during the Lockdown whereas the respective percentage for the Pre-Lockdown period was 78.6%, an observation that is in support of the notion that overall good disease control occurred. The observation that patients with asthma during the Lockdown period were younger may indicate that elderly patients were avoiding hospitals due to fear of COVID-19.

It is worth adding that the lockdown in Athens was imposed during the months that asthmatic exacerbations are expected to rise due to the seasonal increase in allergic burden. Specifically, parietaria has pollination period from early March till mid-June, grasses from early April till mid-June and olive trees during May, depending on the meteorological parameters. It can be easily speculated that “staying at home” during lockdown period (March-April) resulted in allergic patients having lesser exposure to parietaria and grasses causing less allergic rhinitis and allergic asthma symptoms. During the Post-Lockdown period (May), however, people spent most time outside, increasing their exposure to allergens, at the same time that olive pollination was added to that of parietaria’s and grasses’, increasing the total allergenic burden.

A notable observation of our study was that the majority of air pollutants (NO, NO_2_, PM2.5, PM10, CO, SO_2_, benzene) decreased during the quarantine with the exception of O_3_ that increased. Similar findings concerning reduction in air pollution have been reported from previous studies [32,33] during COVID-19 period and the pre-COVID-19 era. Namely, exposure to air pollutants results in airway hyper-responsiveness and remodeling (especially SO_2_), oxidative stress, and decrements in lung function, leading to exacerbation of existing asthma and COPD [34,35]. Pollutants such as NO, NO_2_, PM2.5, PM10, CO and benzene are well described as mainly traffic-related air pollutants [36,37,38]. The reduction of most air pollutants in our study could be reasonably explained via mitigated traffic mobility due to tele-working and social restriction. On the other hand, O_3_ increase can be explained by its seasonal pattern. It is observed that due to the presence of sunlight, O_3_ is higher in spring (Lockdown period) and lower in winter (Pre-Lockdown period) [39,40].

Beyond the observed reduced air pollutants concentrations during the Lockdown, we further managed to correlate hospital admissions in 2020 with the concentration of air pollutants. This observed reduction in air pollutants concentrations during the Lockdown was significantly correlated with the decline in the number of hospital admission for obstructive diseases (asthma and/or COPD). Therefore, by improving air quality through traffic reduction the lockdown may have had a positive effect in obstructive lung disease exacerbations. Nevertheless, it could be argued that increased tropospheric ozone, which was observed during the Lockdown, is harmful and leads to airway inflammation or hyper-responsiveness, with subsequent occurrence of asthma or COPD exacerbations [41,42,43]. Since hospital admissions for these conditions were reduced during the quarantine, the increased ozone may have been of small effect compared to the overall reduction in air pollutants. Similar findings have been reported from other cities [32,44,45] as well.

The main limitation of our study is its restriction to a single centre finding However, during Lockdown, a lot of hospitals in Athens were converted into COVID centers. Hence our hospital was one of the few hospitals in the broader Athens area and the only one at the urban center that its Emergency Department remained open to treat and hence admit non-COVID obstructive lung disease exacerbations. Thus, admissions for asthma or COPD exacerbations were expected to rise during that period, as our referenced hospital covered a larger population area. Consequently, the reduction in hospital admission rates may have been underestimated. As asthma and COPD exacerbation share common symptoms with COVID infection, we had ensured that all patients with respiratory symptoms in our hospital were tested for SARS-CoV-2 with PCR. This was not the case in some countries during the first COVID wave where patients revealing symptoms of cough and shortness of breath are classified as COVID-19 without molecular testing. Thus, in our study we accurately removed the bias of categorizing an obstructive lung disease exacerbation as a COVID-19 case [7].

Since our study deals only with severe exacerbations that need hospitalization, we cannot provide evidence about less severe exacerbations, treated in community. Indeed exacerbations that did not reach the hospital could also have occurred without being recorded.

There is evidence about good overall asthma control without increase in the frequency of community treated exacerbations [26,27,28]. Concerning COPD exacerbations, there is a single center UK based study that exhibits an increase in community treated exacerbations at the beginning of the lockdown, with concomitant decrease in hospitalizations. The increase in community treated exacerbations could result from “behavioral” changes of either patients (i.e., anxiety, fear of hospitalization) or doctors (telephone consultation instead of physical examination) [46]. More specifically in Athens, some patients could have undergone self-treatment or received outpatient treatment. These patients may have sought hospital care just after the lifting of the quarantine and this could explain the increased total admissions observed during the Post-Lockdown period of our study.

Another limitation of our study is that we could not attribute the observed reductions of exacerbations to each parameter separately (reduced exposure to infection related causes of exacerbations, social distancing, universal masking, medical care avoidance behavior, reduction of air pollution). However, we managed to accumulate data on air quality as well as patient population characteristics and perform meaningful associations.

## 5. Conclusions

Lockdown in Athens Greece resulted in a significant reduction in admissions for asthma and COPD, although it coincided with typically expected spring increase due to spring allergens. This reduction correlated significantly with the decline in air pollution, implying that better air quality, probably due to reduction in traffic, is a significant contributor. Moreover, reduced exposure to infection related causes of exacerbations seems to be an important factor for reduced exacerbations. Finally, medical care avoidance behavior among elderly patients with asthma may also have occurred. Asthmatic exacerbations were less severe, whereas COPD exacerbation didn’t differ in severity, but had higher eosinophils at admission possibly implying a non-infectious cause of exacerbation. Overall, patients’ characteristics during the Lockdown indicate good adherence. Our findings help understanding the untold impact of the pandemic on diseases beyond COVID-19.

## Figures and Tables

**Figure 1 jpm-11-00867-f001:**
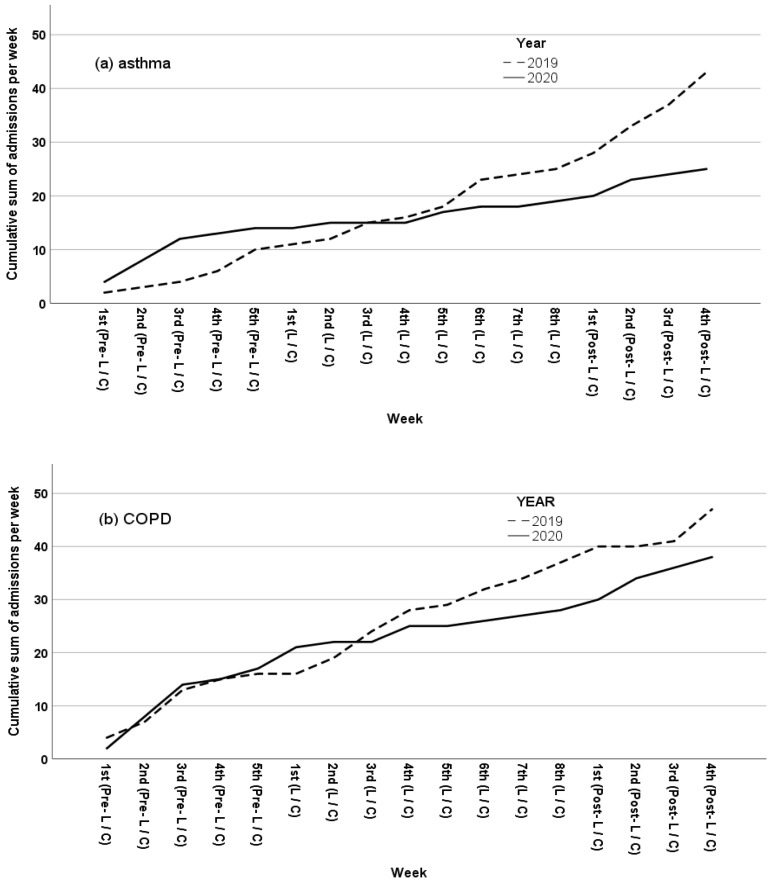
(**a**–**c**) Cumulative sum of admissions per week for 2020 and 2019 for asthma, COPD and total admissions respectively. L = Lockdown; C = Control.

**Figure 2 jpm-11-00867-f002:**
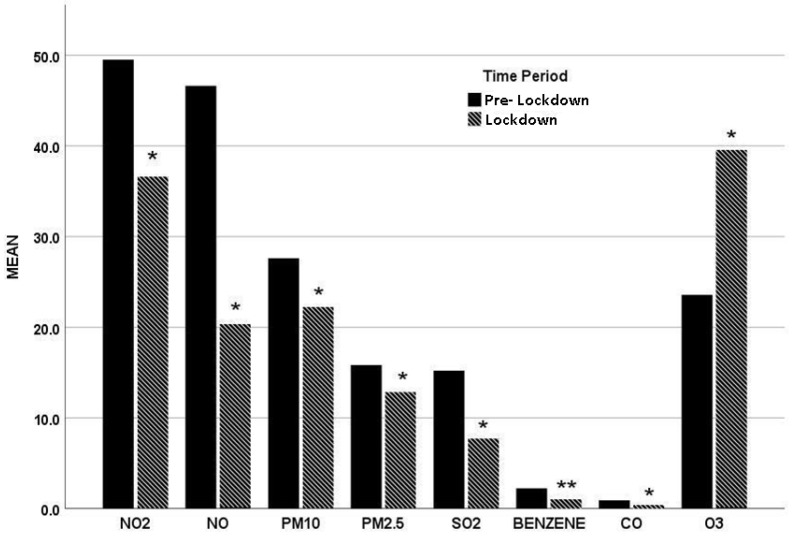
Mean air pollutants concentration in Lockdown and Pre- Lockdown period. All air pollutants are measured in μg/m³ while CO is measured in mg/m³. One star (*) represents statistical significance at a level of 0.05, whereas two stars (**) at a level of 0.01.

**Table 1 jpm-11-00867-t001:** Characteristics of asthma and COPD hospital admissions (*n =* 153).

	Missing (Total)	Asthma (68 Admissions)	COPD (85 Admissions)	*p*-Value
Age (years)	1	57 (44.25–72)	70 (63–78.75)	**<0.001**
Females (%)	0	55.9	37.6	**0.024**
Hospitalization (days)	0	5 (3–8)	7 (5–12)	**<0.001**
Symptom duration before admission (days)	3	4.5 (2–7)	5 (2–7)	0.986
Current Smoker (%)	4	35.8	62.2	**0.001**
Dyspnea (%)	1	86.8	95.2	0.063
Cough (%)	1	70.6	70.2	0.962
Sputum (%)	1	47.1	64.3	**0.033**
Fever (%)	2	27.9	37.3	0.222
Lethargy/Coma (%)	2	8.8	20.5	**0.047**
Respiratory Failure (%)	2	63.2	94	**<0.001**
Hypercapnia (%)	2	20.6	51.8	**<0.001**
Abnormal X-RAY (%)	1	35.3	36.9	0.837
White Blood Cells (admission)	0	10,090 (7537.5–12,532.5)	10,460 (7920–12,550)	0.645
Neutrophiles (%)	0	72.25 (61.275–86)	77.6 (69–87)	**0.047**
Lymphocytes (%)	0	17 (9.075–25.825)	14 (7.5–18.65)	**0.032**
Eosinophiles (%)	0	1.2277 (0.1945–3.4631)	0.4564 (0.0838–1.9037)	**0.010**
Eosinophile (n)	0	120 (20–277.5)	40 (10–175)	**0.008**
Serum CRP (mg/dL)	0	1.7 (0.4–5.175)	2.6 (0.8–11.1)	**0.025**
Intubation/Death(%)	0	5.9	16.5	**0.043**
Flu positive (%)	0	4.4	3.5	1.000

Continuous values are presented as Median (IQR). Statistically significant *p*-values are presented with bold characters. Abbreviations: COPD = Chronic Obstructive Pulmonary Disease, pts = patients, *n =* number.

**Table 2 jpm-11-00867-t002:** Comparisons among incidence rates between lockdown and the different study periods.

	Asthma			COPD			Total Admissions		
	IR	IRR with 95% CI	*p*	IR	IRR with 95% CI	*p*	IR	IRR with 95% CI	*p*-Value
Lockdown 2020 (Reference)	0.625	1	-	1.375	1	-	2	1	-
Pre-lockdown	2.8	4.48 (1.6–12.4)	*p* = 0.004	3.4	2.5 (1.2–5.3)	*p* = 0.019	6.2	3.1 (1.7–5.7)	*p <* 0.001
Post-lockdown	1.5	2.4 (0.7–7.9)	*p* = 0.148	2.5	1.8 (0.8–4.3)	*p* = 0.171	4	2.0 (1.0–4.0)	*p* = 0.05
Pre-Control	2	3.2 (1.1–9.4)	*p* = 0.034	3.2	2.3 (1.1–5.0)	*p* = 0.031	5.2	2.6 (1.4–4.8)	*p* = 0.003
Control 2019	1.875	3 (1.1–8.3)	*p* = 0.033	2.625	1.9 (0.9–4.0)	*p* = 0.082	4.5	2.3 (1.2–4.1)	*p* = 0.007
Post-Control	4.5	7.2 (2.7–19.4)	*p <* 0.001	2.5	1.8 (0.8–4.3)	*p* = 0.171	7	3.5 (1.9–6.5)	*p <* 0.001

IR = Incidence Rate per week, IRR = Incidence Rate Ratio (between the respective period and lockdown); IRR with 95% CI are rounded to 1 decimal place, *p = p*-value; statistically significant *p*-values are presented with bold characters.

**Table 3 jpm-11-00867-t003:** Comparisons in admissions between Pre-Lockdown and Lockdown.

	Asthma			COPD		
	Pre-Lockdown (*n =* 14)	Lockdown (*n =* 5)	*p*-Value	Pre-Lockdown (*n =* 17)	Lockdown (*n =* 11)	*p*-Value
Age (years)	63.5 (55.5–81.5)	50 (38–59.5)	0.046	68 (63.5–75)	76 (60–83)	0.621
Females (%)	71.4	60	1.000	29.4	18.2	0.668
Hospitalization (days)	6 (2–8)	4 (2.5–13.5)	0.852	7 (5.5–13)	9 (6–14)	0.569
Symptom duration before admission (days)	4 (2–7)	7 (4–10.5)	0.138	7 (4–9.25)	5 (2–7)	0.278
Current Smoker (%)	57.1	20	0.303	68.8	45.5	0.264
Dyspnea (%)	85.7	80	1.000	100	100	-
Cough (%)	71.4	60	1.000	88.2	72.7	0.353
Sputum (%)	57.1	40	0.628	88.2	63.6	0.174
Fever (%)	35.7	20	1.000	43.8	27.3	0.448
Lethargy/Coma (%)	14.3	0	1.000	18.8	36.4	0.391
Respiratory Failure (%)	78.6	20	0.038	87.5	100	0.499
Hypercapnia(%)	28.6	0	0.530	56.3	63.6	1.000
Abnormal X-RAY (%)	42.9	20	0.603	41.2	36.4	1.000
White Blood Cells	9145 (7010–13,242.5)	12,640 (8225–16,485)	0.267	10,500 (8745–15,270)	9270 (7090–14,800)	0.290
Neutrophiles (%)	73.4 (57.8–77.1)	72 (57.9–85.3)	0.853	79 (74.3–85.5)	75 (61–87)	0.451
Lymphocytes (%)	17.2 (12.9–31.3)	13.1 (8.6–19.6)	0.195	10.7 (7–16.1)	15 (8–25)	0.437
Eosinophiles (%)	0.4 (0.1–1.7)	2.8 (1–20.1)	0.052	0.1 (0–0.8)	2.2 (0.5–4.2)	0.017
Eosinophile (n)	30 (10–205)	220 (95–3225)	0.070	10 (0–105)	170 (30–300)	0.012
Serum CRP (mg/dL)	3.1 (0.9–7.4)	0.3 (0–13.5)	0.115	3.8 (0.8–16.6)	1.5 (0.2–8.8)	0.110
Intubation/Death (%)	7.1	0	1.000	23.5	18.2	1.000
Flu positive (%)	14.3	0	1.000	5.9	0	1.000

Continuous values are presented as Median (IQR). Statistically significant *p*-values are presented with bold characters. Abbreviations: *n =* number; COPD = Chronic Obstructive Pulmonary Disease. Air Pollution: Comparisons between Lockdown and Pre-Lockdown and Correlation with Hospital Admissions.

**Table 4 jpm-11-00867-t004:** Spearman’s correlation coefficients for air pollutants and admissions for 2020.

		NO	NO_2_	PM2.5	PM10	O_3_	CO	SO_2_	BENZENE
Total admissions	r	**+0.694**	**+0.601**	**+0.526**	+0.435	**−0.635**	**+0.707**	+0.373	**+0.721**
	*p*	0.002	0.011	0.030	0.081	0.006	0.002	0.140	0.001
Asthma admissions	r	+0.401	+0.311	**+0.543**	+0.358	−0.368	**+0.526**	**+0.516**	**+0.546**
	*p*	0.111	0.224	0.024	0.158	0.146	0.030	0.034	0.023
COPD admissions	r	**+0.642**	**+0.554**	**+0.492**	+0.439	**−0.56**	**+0.616**	+0.246	**+0.621**
	*p*	0.005	0.021	0.045	0.078	0.019	0.009	0.342	0.008

r = Spearman’s correlation coefficient. Statistically significant correlation (*p*-value < 0.05) is highlighted by Bold r.

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
