# Peer review of "Effect of COVID-19-Related Lockdown οn Hospital Admissions for Asthma and COPD Exacerbations: Associations with Air Pollution and Patient Characteristics"

_jpm, 2021, doi:10.3390/jpm11090867_

Round 1
Reviewer 1 Report
The manuscript “Effect of COVID-19-related lockdown οn hospital admissions for asthma and COPD exacerbations: associations with air pollution and patient characteristics” evaluates the number of asthma and CODP exacerbation before, during and after Quarantine in Grece in 2020 and correlates them with air pollution data and patients’ characteristics. The study is interesting, provides new information about the impact of viral and environmental factors on obstructive lung diseases course.
Minor remarks:
- The English proofreading by native spiker is recommended for this manuscript.
- The study hypothesis should be formulated more carefully and be focused on the main study goal (environmental and viral impact on asthma and CODP exacerbations).
- Please revise the abbreviations in the text many of should be corrected, also the abbreviation should be explained in first place used.
- p-value instead of p should be placed in tables’ heading.
Author Response
RESPONSE TO REVIEWERS COMMENTS
1. Comment:The English proofreading by native speaker is recommended for this manuscript.
Response: Thank you for this comment. The text was revised form a native English speaker
- Comment: The study hypothesis should be formulated more carefully and be focused on the main study goal (environmental and viral impact on asthma and COPD exacerbations).
Response: Thank you for your suggetsion. We have formulated carefully the hypothesis and focused on the main stidy goal as suggestd and corrected the text to read:
"We thus hypothesize that reduced patients' hospitalisation could be hinged reduction in air pollutants from and reduced exposure to infection causes of exacerbations."
- Comment: Please revise the abbreviations in the text many of should be corrected, also the abbreviation should be explained in first place used.
Response: Thank you for this comment, we have corrected the text so as the abbreviations are explained in the first place used
- Comment: p-value instead of p should be placed in tables’ heading.
Response: Thank you for this comment we have corrected the text to read p-value
Reviewer 2 Report
I thought this was a worthwhile and useful study but there are a couple of changes which I think should be addressed before publication.
- Abstract: I think the authors need to consider the impact of social distancing and isolating at home on exposure to infectious causes of exacerbations. In some countries patients with COPD and severe asthma were required to isolate at home and were advised not to leave the house during lockdown. Did this happen in Greece? With reduction in exposure to infection you might expect a reduction in exacerbations.
- Introduction, first paragraph: as above - I think the authors need to comment on reduced exposure to infections as well as environmental triggers in lockdown.
- Results: Could the correlations be presented as scatterplots please?
- Discussion: Again - I think reduced infection exposure should be discussed as one of the factors that might be associated with reduced exacerbations.
- Line 343 should read 'beginning of lockdown'
- Line 374: please remove the phrase 'Please add:'
Author Response
- Comment: Abstract -: I think the authors need to consider the impact of social distancing and isolating at home on exposure to infectious causes of exacerbations. In some countries patients with COPD and severe asthma were required to isolate at home and were advised not to leave the house during lockdown. Did this happen in Greece? With reduction in exposure to infection you might expect a reduction in exacerbations.
Response: Thank you for this comment. We have included the effect of social distancing and isolating at home on exposure that decreased the infectious causes of exacerbations in the abstract, the discussion and the conclusion. In Greece during the first lockdown obstructive lung disease patients were adhernt to the quarantine and therefore social distancing and isolating at home lead to reduced infectious causes of exacerbations.
- Comment: Introduction- first paragraph: as above - I think the authors need to comment on reduced exposure to infections as well as environmental triggers in lockdown.
Response: Thank you for this comment and helping us ameriorate our manuscript. We have included the effect of decreased infectious causes of exacerbations and environmental triggers in the abstract, the discussion and the conclusion and commented extensively in the discussion.
- Comment: Results: Could the correlations be presented as scatterplots please?
Response: Thank you for your suggestion. We have presented the correlations as scatterplots as requested and added them as supplement data (S1):
- Discussion: Again - I think reduced infection exposure should be discussed as one of the factors that might be associated with reduced exacerbations:
Response: Thank you for highlighting the above issue. Indeed reduced infection exposure might be associated with reduced exacerbations. This has been added in the discussion and clarified as previously it was implied as the effect of social distancing and isolating at home.
- Comment: Line 343 should read 'beginning of lockdown'. Response: Thank you for this comment we have corrected the text accordingly
- Comment: Line 374: please remove the phrase 'Please add:'
Response: Thank you for the comment, we removed it as suggested
